# SynopGround: A Large-Scale Dataset for Multi-Paragraph Video Grounding from TV Dramas and Synopses

Chaolei Tan[*][†]
Sun Yat-sen University
Guangzhou, China
tanchlei@mail2.sysu.edu.cn

Zihang Lin[*][†]
Sun Yat-sen University
Guangzhou, China
linzh59@mail2.sysu.edu.cn

Junfu Pu
ARC Lab, Tencent PCG
Shenzhen, China
jevinpu@tencent.com

Zhongang Qi
ARC Lab, Tencent PCG
Shenzhen, China
zhongangqi@tencent.com

Wei-Yi Pei
Tencent Video, PCG
Shenzhen, China
weiyipei@tencent.com

Zhi Qu
Tencent Video, PCG
Shenzhen, China
jessonqu@tencent.com

Yexin Wang
Tencent Video, PCG
Shenzhen, China
yexinwang@tencent.com

Ying Shan
ARC Lab, Tencent PCG
Shenzhen, China
yingsshan@tencent.com

Wei-Shi Zheng
Sun Yat-sen University, Key
Laboratory of Machine
Intelligence and Advanced
Computing, Ministry of
Education
Guangzhou, China
wszheng@ieee.org

Jian-Fang Hu[1][‡]
Sun Yat-sen University,
Guangdong Province Key
Laboratory of Information
Security Technology
Guangzhou, China
hujf5@mail.sysu.edu.cn

## Abstract

Video grounding is a fundamental problem in multimodal content understanding, aiming to localize specific natural language queries in an untrimmed video. However, current video grounding datasets merely focus on simple events and are either limited to shorter videos or brief sentences, which hinders the model from evolving toward stronger multimodal understanding capabilities. To address these limitations, we present a large-scale video grounding dataset named SynopGround, in which more than 2800 hours of videos are sourced from popular TV dramas and are paired with accurately localized human-written synopses. Each paragraph in the synopsis serves as a language query and is manually annotated with precise temporal boundaries in the long video. These paragraph queries are tightly correlated to each other and contain a wealth of abstract expressions summarizing video storylines and specific descriptions portraying event details, which enables the model to learn multi-modal perception on more intricate concepts over longer context dependencies. Based on the dataset, we further introduce a more complex setting of video grounding dubbed Multi-Paragraph Video Grounding (MPVG), which takes as input multiple paragraphs and a long video for grounding each paragraph query to its temporal

interval. In addition, we propose a novel Local-Global Multimodal Reasoner (LGMR) to explicitly model the local-global structures of long-term multimodal inputs for MPVG. Our method provides an effective baseline solution to the multi-paragraph video grounding problem. Extensive experiments verify the proposed model's effectiveness as well as its superiority in long-term multi-paragraph video grounding over prior state-of-the-arts. Dataset and code are publicly available. Project page: https://synopground.github.io/.

## CCS Concepts

• **Information systems** → **Multimedia and multimodal retrieval**; **Video search**; **Novelty in information retrieval**.

## Keywords

Vision and Language, Long-Term Multimodal Content Understanding, Multi-Paragraph Video Grounding, Large-Scale Dataset

**ACM Reference Format:**
Chaolei Tan, Zihang Lin, Junfu Pu, Zhongang Qi, Wei-Yi Pei, Zhi Qu, Yexin Wang, Ying Shan, Wei-Shi Zheng, Jian-Fang Hu. 2024. SynopGround: A Large-Scale Dataset for Multi-Paragraph Video Grounding from TV Dramas and Synopses. In *Proceedings of the 32nd ACM International Conference on Multimedia (MM '24), October 28-November 1, 2024, Melbourne, VIC, Australia.* ACM, New York, NY, USA, 10 pages. https://doi.org/10.1145/3664647.3681042

---

[*]Equal contribution.

[†]Work done during an internship at ARC Lab, Tencent PCG.

[‡]Corresponding author.

*MM '24, October 28-November 1, 2024, Melbourne, VIC, Australia*
© 2024 Copyright held by the owner/author(s).
ACM ISBN 979-8-4007-0686-8/24/10
https://doi.org/10.1145/3664647.3681042

## 1 Introduction

As a crucial problem in multimodal understanding, video grounding aims at linking semantically relevant temporal intervals in an untrimmed video with specific natural language queries. Recently, video grounding has received increasing attention since a wide range of downstream applications can be promoted by it, such as improving the searching granularity of video retrieval [4, 13, 16, 22, 77], enabling language-aware scenarios of video editing [6, 20, 21, 36],

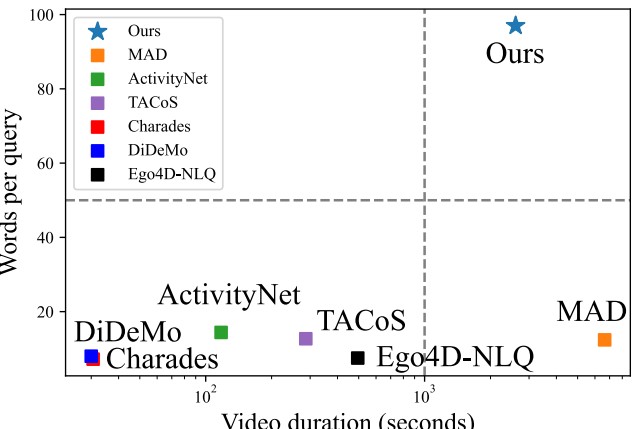

**Figure 1: Comparison of video grounding datasets. Our Syn-opGround is the first dataset to introduce the challenges of both long videos and long queries into video grounding.**

and making video question answering [32, 39, 41, 44, 45, 78, 83] more evidence-based. Up to now, a large number of datasets and methods have been established to advance this line of research.

However, considerable drawbacks still exist in previous video grounding datasets [2, 23, 24, 37, 60, 66]. First of all, as presented in Figure 1, most of commonly-used datasets are constructed upon short videos and brief sentence queries. This setup limits the model in developing stronger abilities that can model and bridge the long-form videos [27, 57] and long-text queries [23, 81]. Besides, shorter queries that describe detailed events (as shown in Table 1), are more prone to causing the risk of semantic ambiguity in referring expressions [52, 56], i.e., the occurrence of one-to-many correspondence between queries and moments, which will adversely affect the model learning. In particular, this ambiguity issue is more prominent for the recently proposed MAD [66] dataset which features long input videos but short general descriptions. For example, it is highly likely to find cases where multiple moments are semantically corresponding to the same short description like "She steps closer." (shown in Table 1), especially when searching content in a long video. Furthermore, as listed in Table 1, existing benchmarks are tailored for language queries referring to low-level visible activities, while all of them overlook the importance of more complex events and abstract concepts. Such drawbacks actually limit the applications of video grounding in scenarios where complex descriptions with abstract concepts should be associated with long-term videos. For example, accelerating the movie post-production by automatically integrating raw footage into a coherent story based on the plot scripts is a practical need, but it cannot be satisfied by the current video grounding techniques developed from existing datasets.

In this work, we curate and present a large-scale dataset called SynopGround to address the current limitations of video grounding datasets. We collect and manually annotate episodes from popular TV dramas of various genres, yielding a large-scale video grounding dataset consisting of over 2800 hours of fully-annotated videos. Specifically, for each video, we crawl its human-written synopsis consisting of multiple paragraphs from the Internet, and further annotate the precise temporal boundaries for each paragraph in the given synopsis. As demonstrated in Figure 1, our dataset has both

**Table 1: Comparison of queries in different datasets. The red-bold text is a detailed description, while the blue-italic text is an abstract and concise expression.**

| Dataset | Query |
|---|---|
| Charades [23] | A person runs to the window then looks out. |
| DiDeMo [2] | The little girl jumps back up after falling. |
| TACoS [60] | He flips the eggs, making an omelet. |
| ActivityNet[37] | A woman walks to the piano and briefly talks to the elder man. |
| Ego4d-NLQ [24] | What did I pick from the fridge? |
| MAD [66] | She steps closer. |
| SynopGround (Ours) | …Stefan and Elena decided to **go to the cabin** left by Elena's parents, where they *spent a happy time.* … |

significantly longer average video length and average query length than most existing ones. It is the first video grounding dataset that can support the research on long-term contextual video grounding with complex queries. Moreover, compared to the short sentence queries in the existing datasets, our long paragraph queries can unambiguously indicate one-to-one correspondence between language queries and target moments, which is crucial for learning accurate cross-modal alignment. Furthermore, as shown in Table 1, there are very concrete descriptions for visible activity concepts like "go to the cabin", as well as extremely concise and abstract expressions like "spent a happy time" in the query from our dataset. This enables to learn and evaluate the comprehensive understanding of semantic concepts at diverse abstraction levels.

Based on our dataset, we pioneer to introduce and explore a more challenging and complex setting of video grounding called Multi-Paragraph Video Grounding (MPVG). The MPVG task receives a multi-paragraph synopsis and a long narrative video as inputs to localize the temporal interval of each synopsis paragraph from the video. To promote and inspire future research, we further propose a novel Local-Global Multimodal Reasoner (LGMR) to explicitly model the local-global structures of long-term multimodal inputs and conduct iterative cross-modal reasoning within and across the two levels of structures for effectively tackling the multi-paragraph video grounding problem. Extensive experiments demonstrate the effectiveness of our baseline in the proposed research direction.

The main contributions of this work are summarized as follows:

- We present SynopGround, a large-scale video grounding dataset consisting of over 2800 hours of TV drama videos with manual temporally-annotated professional synopses.
- Based on the dataset, we first introduce a challenging Multi-Paragraph Video Grounding (MPVG) task and propose a novel Local-Global Multimodal Reasoner (LGMR) baseline.
- We are the first to incorporate long-form videos and long abstract paragraphs into video grounding. Comparison results show the unique advantages of our dataset and the efficacy of our baseline in multi-paragraph video grounding.

## 2 Related Work

In this section, we aim to review and discuss the existing works in the video grounding and narrative video understanding areas.

### 2.1 Video Grounding

**Datasets.** In video grounding, Charades-STA [23], DiDeMo [2], ActivityNet-Captions [37] and TACoS [60] are the four most commonly used datasets for model training and evaluation. However, a

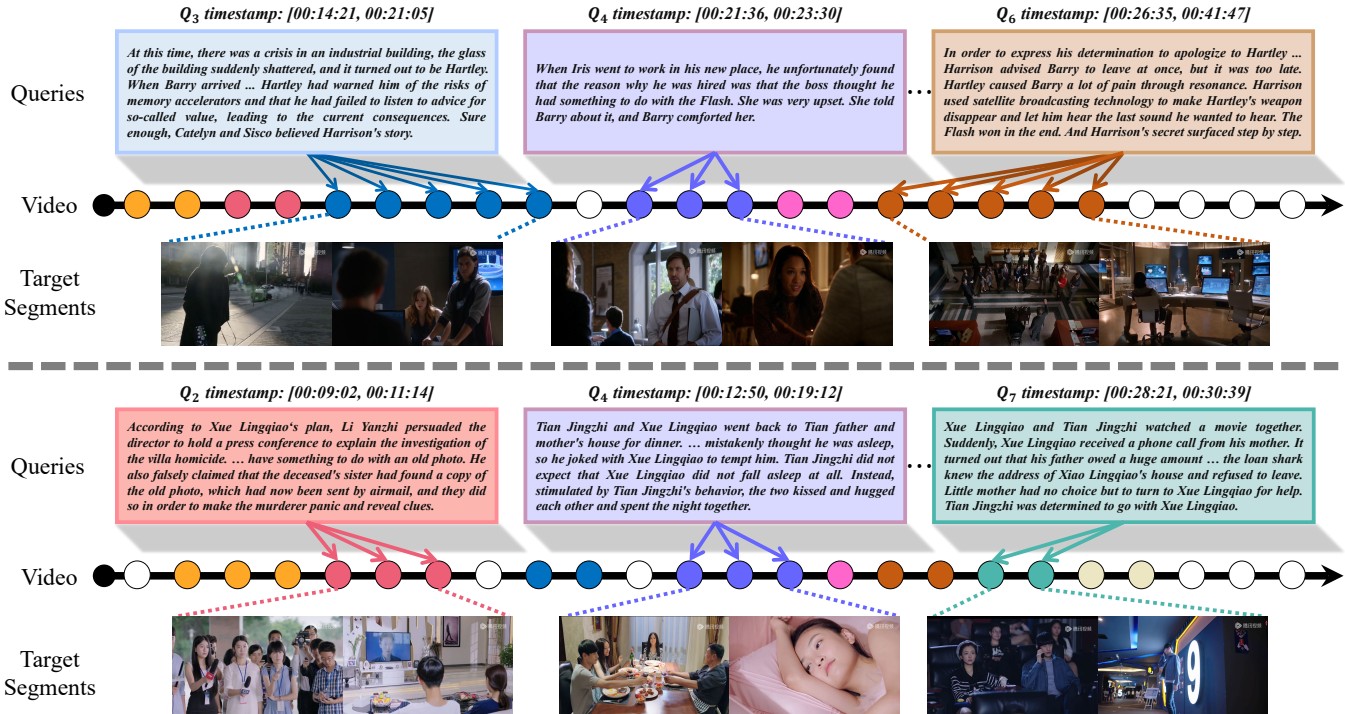

**Figure 2: Illustration of the proposed Multi-Paragraph Video Grounding (MPVG) problem and two representative samples in our SynopGround. Given a video and a synopsis $Q$ that contains $N$ paragraphs $\{Q_1, Q_2, ..., Q_N\}$, the model should predict the corresponding temporal interval for each paragraph $Q_i$ in the form of starting and ending time.**

majority of these datasets [23, 37, 60] are adapted from pre-existing video datasets tailored for closed-set recognition or localization tasks [7, 62, 64], which makes them severely limited to a pre-defined set of visual and linguistic concepts. DiDeMo [2] is a customized video grounding dataset. However, it overly simplifies the annotation and only supports the model to select from 5 evenly-divided segments of the video. In addition, shortcut learning issues caused by distribution biases in previous datasets have been reported [38, 56], which could adversely affect the benchmark reliability. Moreover, the above datasets [2, 23, 37, 60] are all constructed on a relatively small-scale collection of short videos and simple sentence descriptions, which cannot support the need of large-scale model training for long-term contextual video-language understanding that incorporates complex language queries. Recently, the Ego4d-NLQ [24] and MAD [66] datasets are introduced. Nevertheless, both of them still focus on the simple visible activities and short-term temporal events. Specifically, Ego4d-NLQ contains egocentric videos and adopts brief interrogative queries asking about simple visible fact grounded on a short video interval. The MAD dataset is semi-automatically constructed on movies with audio descriptions and its average video length is significantly longer compared to the other existing datasets. However, the language queries of MAD are still brief sentences that individually describe short-term events in the long video. Different from all of the prior works, our proposed SynopGround is the first video grounding dataset that considers both long-form videos and long-text queries. Additionally, we adopt narrative videos conveying storylines and tightly correlated synopsis paragraphs as inputs, which poses more challenges for the

video grounding model to understand high-level story plots and invisible abstract concepts over a longer context.

**Tasks and settings.** Early research of video grounding has largely focused on grounding single sentences in videos, i.e., the Video Sentence Grounding (VSG) task introduced by Gao et al. [23] and Hendricks et al. [2]. Afterwards, a series of extended tasks [5, 18, 40, 42] have been proposed. Escorcia et al. [18] first introduced the task of Video Moment Corpus Retrieval (VCMR) for combining video retrieval and moment localization, and Lei et al. [42] curated the TVR dataset to incorporate multi-modal information into VCMR. Lei et al. [40] proposed QVHighlights dataset along with a new direction combining highlight detection and moment retrieval. To reduce ambiguity by exploring inter-query context, some recent works [1, 5, 12, 34, 63, 71] have shifted to a multi-query version of video sentence grounding, where the model is required to understand several temporally ordered sentences and localize each sentence in a richer context. Specifically, Bao et al. [5] first studied multi-sentence video grounding in a fully-supervised setting, and the semi-supervised setting [34] as well as weakly-supervised setting [69] have also been investigated after that. These prior works have shown the great potential of contextually understanding multimodal content in untrimmed videos and language descriptions. In this work, we take a step further to introduce a more challenging setting of contextual video grounding called Multi-Paragraph Video Grounding (MPVG). It requires to understand both short-term intra-paragraph semantics and long-term inter-paragraph dependencies, which connects the complex temporal structures of long videos with the complicated semantics of long paragraphs.

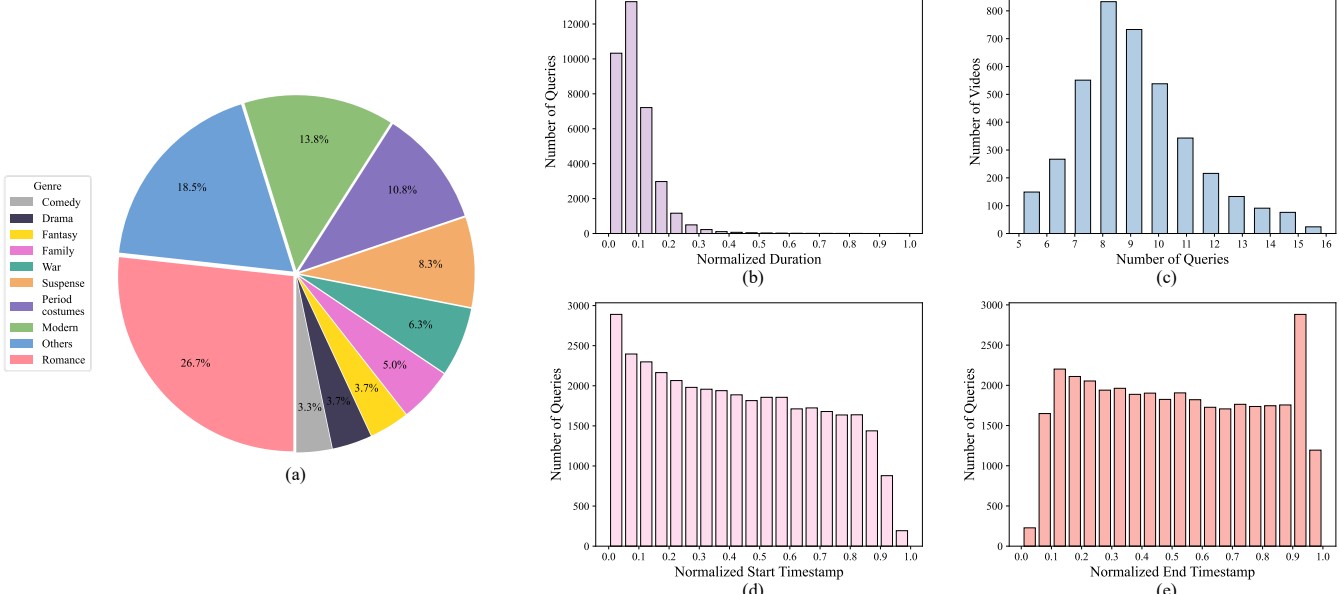

**Figure 3: Data distribution in our dataset. (a): Genre distribution of TV dramas. (b): Normalized duration of target video segments. (c): Number of queries per video. (d): Normalized start timestamp distribution. (e): Normalized end timestamp distribution.**

**Methods.** A lot of approaches [11, 14, 28, 31, 43, 47–49, 51, 53, 55, 68, 70, 74–76, 79, 82, 84–87, 89, 90] have been developed for video grounding over recent years. As summarized in [88], these methods can be roughly categorized into proposal-based and proposal-free methods. Proposal-based methods typically involve a two-stage process of generating moment proposals for relevance score ranking, which often leads to issues like inefficiency and limited adaptability. In contrast, proposal-free methods tend to have better efficiency and they directly predict the temporal boundaries based on cross-modal interactions, which is more suitable for various real-world scenarios. Considering the long-term characteristics in our dataset, we choose to design our Local-Global Multimodal Reasoner (LGMR) in the more efficient proposal-free fashion. Different from the previous approaches that only consider modeling the cross-modal correspondence between a single paragraph and the video, our method is constructed by reasoning through the local and global structures of multiple paragraphs and the video.

## 2.2 Narrative Video Understanding

Understanding visual content presented in narrative videos is an important area with many works [3, 15, 25, 26, 29, 30, 33, 54, 59, 65–67, 72, 80] proposed accordingly. Most of these prior works neglect to model and understand the content of narrative videos based on their high-level storylines while focusing on specific downstream applications, such as movie genre classification [65], character identification [26, 29, 54], action localization [25], scene segmentation [15, 30], and shot classification [59]. In addition to that, some works have started pursuing story-level understanding in many different ways. For instance, Tapaswi et al. [72] proposed MovieQA dataset to comprehend movie stories by question-answering. MSA [80], CMD [3], and SyMoN [67] datasets utilized synopses as language queries and formulated movie understanding

as text-to-video retrieval. This line of research is closely related to ours. However, we focus on a more challenging video grounding task with long contexts and complex queries, in which the model should understand the long-range cross-modal dependencies so as to reason about the video grounding results at story level.

## 3 SynopGround Dataset

In this section, our goal is to give a formal definition of our introduced multi-paragraph video grounding problem and illustrate the details of the data collection, annotation, statistics, and processing.

## 3.1 Problem Formulation

Considering video paragraph grounding [5] is limited to a multi-query version of short single sentence grounding, we introduce a more challenging setting to incorporate long abstract paragraphs as queries called Multi-Paragraph Video Grounding (MPVG). Specifically, given an untrimmed video $\mathcal{V}$ and $N$ consecutive paragraph queries $Q = \{Q_1, Q_2, ..., Q_N\}$ as input, the output should be $N$ temporal intervals $\{\mathcal{T}_1, \mathcal{T}_2, ..., \mathcal{T}_N\}$ corresponding to each of the paragraph queries, where $\mathcal{T}_i = \left(t_s^{(i)}, t_e^{(i)}\right)$ indicates the starting timestamp $t_s^{(i)}$ and ending timestamp $t_e^{(i)}$ for the $i$-th paragraph query in the target video. In our dataset, the video $\mathcal{V}$ is an episode from a TV drama, and $Q$ is the corresponding human-written synopsis that contains $N$ paragraphs, with $Q_i$ indicating the $i$-th paragraph in the synopsis $Q$. Illustration of our MPVG is in Figure 2.

## 3.2 Data Collection and Annotation

We collect all the TV drama episodes from a leading online platform Tencent Video with official acknowledgement and permission. The plot synopsis for each episode of the TV dramas is scraped from

**Table 2: Detailed comparison with existing video grounding datasets. Our SynopGround is at a larger scale in terms of the total duration of videos and it contains precise temporal annotations generated by human annotators. It is also the first large-scale dataset that considers both long-form videos and long-text queries for multi-paragraph video grounding.**

| Dataset | Charades-STA[23] | ANet-Captions[37] | DiDeMo[2] | TACoS[60] | Ego4d-NLQ[24] | MAD[66] | **SynopGround** |
|---|---|---|---|---|---|---|---|
| Domain | Indoor | Open | Open | Cooking | Open | Open | Open |
| Annotation Mode | Semi-Automatic | Manual | Manual | Manual | Manual | Semi-Automatic | Manual |
| Paragraph Query | No | No | No | No | No | No | **Yes** |
| # Videos | 6,672 | **14,926** | 10,464 | 127 | 1659 | 650 | 3,987 |
| # Queries | 16,124 | 71,953 | 40,543 | 18,818 | 19,170 | **384,600** | 36,002 |
| # Words / Query | 7.2 | 14.4 | 8.0 | 12.7 | 7.5 | 12.4 | **97.0** |
| Duration / Video | 30.6s | 117.6s | 30.0s | 286.6s | 495s | **6,646.0s** | 2,608.4s |
| Duration / Moment | 8.1s | 37.1s | 6.5s | 6.1s | 3.9s | 4.1s | **239.5s** |
| Total Duration | 57.1h | 487.6h | 88.7h | 10.1h | 228.1h | 1,200.0h | **2,884.9h** |

**Table 3: Statistics of dataset division.**

| Data Split | # Dramas | # Videos | # Queries |
|---|---|---|---|
| Training | 470 | 3,187 | 28,677 |
| Validation | 190 | 400 | 3,791 |
| Testing | 192 | 400 | 3,534 |

a specialized TV review website[1] that contains lots of synopses of the most popular TV drama episodes written by professionals. Synopses that are too long or too short are discarded to ensure an adequate number of paragraphs in each synopsis. A total of 520 licensed and high-viewership TV dramas with textual synopses are finally selected to constitute our dataset, and we randomly sample several episodes from each selected TV drama to further annotate. Specifically, annotators are asked to read and understand the synopsis in advance. They then thoroughly watch the corresponding TV drama episode to determine the starting and ending time of the video content depicted by each synopsis paragraph.

Our data annotation pipeline is organized into multiple rounds to ensure the annotation quality. Specifically, all collected videos are first divided into numerous disjoint subsets of videos. In each annotation round, synopsis paragraphs for videos in one subset will be annotated with timestamps and each annotator is told to provide a score to indicate level of confidence in the annotated results. Afterwards, we first discard samples with low confidence as an initial cleanup, and then some of the remaining samples are selected to be manually checked in terms of quality. If the annotation quality is thought of as satisfactory, the annotation process will move on to another unlabeled subset of video data. Otherwise, the current batch of data would be re-annotated. The above procedures are repeated by tens of annotators until we finish the annotation of all candidate samples. For post-annotation assessment, we randomly select a proportion of the annotated data to be re-annotated by other annotators. Concretely, we calculate the temporal IoU (Intersection over Union) between the two results from different annotators, which reaches a value of about 85%. This assessment result is much better than those of other datasets, such as the ActivityNet [37], where different annotators only achieve an agreement degree around 70%. The higher degree of agreement across different

[1]URL: https://www.tvmao.com. All texts are translated into English using Tencent Cloud Translation with full permission and compliance.

annotators in our dataset verifies the effectiveness of our designed pipeline for data annotation and quality control.

### 3.3 Data Statistics

**Data distribution.** We first illustrate some statistical distributions of our dataset in Figure 3. As shown in Figure 3 (a), the TV dramas used in our dataset cover a wide spectrum of genres, which demonstrates the diversity of the collected data. In Figure 3 (b), we show the normalized duration of the target video segments. Most of the target video segments cover less than 20% of the full episode, which can be challenging for the model to correctly localize. In Figure 3 (c), we visualize the distribution of the number of queries/paragraphs in each synopsis, and most synopses are composed of 5-13 paragraphs. Exploring the contextual information among these paragraphs is important for achieving promising performance in our multi-paragraph video grounding task. In Figure 3 (d) and (e), we visualize the temporal distributions of the starting timestamps and ending timestamps of the target video segments. Both of them approximately present a uniform distribution, which ensures the model cannot benefit much from the distribution bias.

**Detailed comparison with other datasets.** In Table 2, we compare our dataset with other existing datasets in detail. As suggested, our videos are much longer in duration than those of Charades-STA [23], ActivityNet-Captions [37], DiDeMo [2], TACoS [60] and Ego4d-NLQ [24]. Although the average video duration in our dataset is shorter than that of MAD, our total duration of videos is more than twice that of MAD, showing that our dataset is at a larger scale. Furthermore, the duration of target segments in our dataset is significantly longer while the normalized target span is still short, making our target moments challenging to be localized. Note that some datasets like MAD have shorter normalized target span than ours, but their short and general queries bring the harmful and undesirable semantic ambiguity issue as mentioned before. In addition, our dataset is the first to incorporate paragraph queries, and the average number of words in each query is significantly larger than those of other datasets, which greatly reduces the semantic ambiguity of the queries. Moreover, our synopsis queries involve both abstract expressions and concrete descriptions, enabling the model to learn semantic concepts at more diverse abstraction levels.

**Data splits.** As shown by the statistics in Table 3, we carefully divide the entire data into three non-overlapping splits for training,

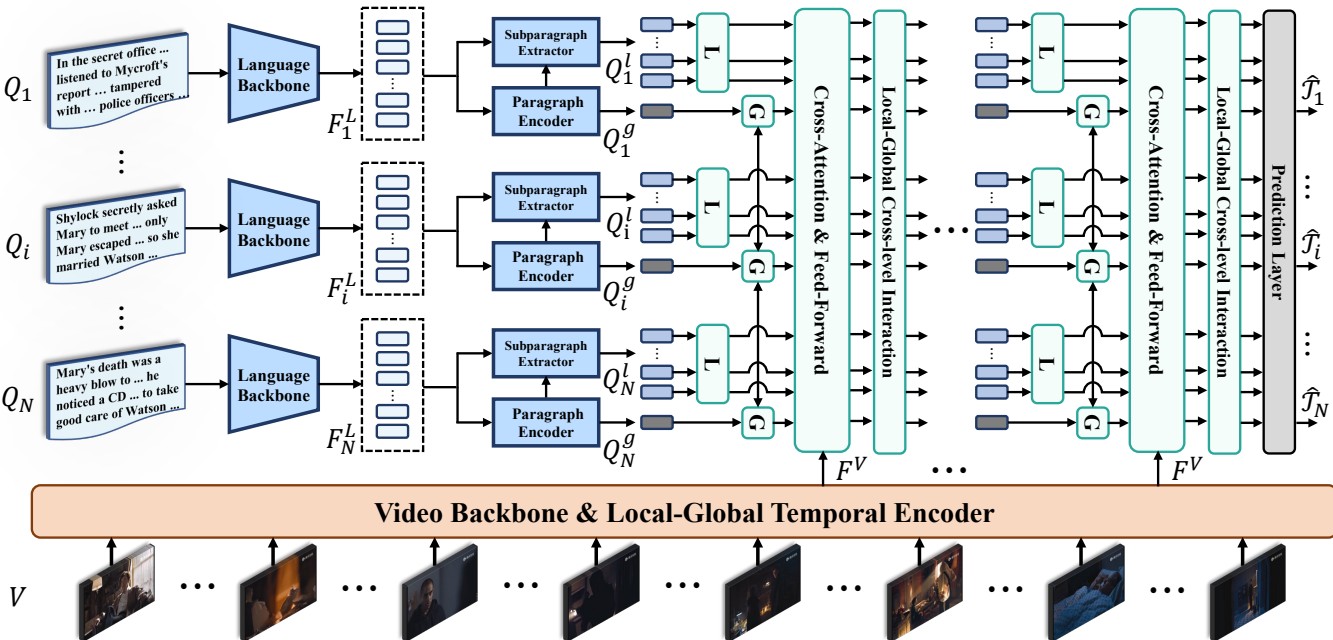

**Figure 4: Our proposed Local-Global Multimodal Reasoner (LGMR). It consists of a local-global temporal encoder for structural long-term temporal modeling and a local-global iterative decoder to adaptively reason through local and global semantics.**

validation, and testing. Our training, validation, and testing sets consist of 3,187, 400, and 400 videos, respectively. It is worth noting that each video is an episode from a TV drama. Additionally, we manually guarantee half of the videos for validation/testing are sourced from TV dramas that are not selected for the training set.

### 3.4 Data Pre-processing

To promote data utilization, we provide pre-extracted features for public release. Specifically, pre-trained CLIP [58] ViT-L/14 model is adopted to extract frame features for videos using a sampling rate of 3 FPS. Additionally, we extract segment features for videos with SlowFast [19] network, which is pre-trained on Kinectics-600 [9, 10] and AVA [25] datasets. To capture the character and dialogue information related to the storylines, we extract embedded subtitles using OCR models DBNet [46] and SVTR [17]. Then, we encode each extracted subtitle to a feature representation using a pre-trained RoBERTa [50] model. The pre-extracted CLIP, SlowFast, and OCR features describe the video from different aspects. They provide complementary information of the video and are beneficial for aligning synopsis with the video. Due to copyright restrictions, we cannot release the raw video frames but we will provide URL links where researchers can access and view the original videos.

## 4 Method

In this section, we illustrate the details of our proposed baseline method to tackle the multi-paragraph video grounding problem.

### 4.1 Overview

As shown in Figure 4, our proposed Local-Global Multimodal Reasoner (LGMR) consists of a local-global temporal encoder for encoding the long input videos and a local-global iterative decoder

for decoding the long paragraph queries. The video encoder decomposes the temporal correlations of long videos into intra-window and inter-window parts for efficient long-term temporal modeling. The query decoder first extracts subparagraph representations with a set of learnable queries guided by the global semantics of paragraphs, and then repeatedly conducts cross-modal reasoning within and across the local and global queries. We elaborate on more architectural details in the following.

### 4.2 Local-Global Temporal Encoder

Given the long-form video inputs, we design a local-global attentive encoder to capture the evolving temporal dynamics of long-term video content, which exploits more structural temporal information than straightforward full attention. Specifically, we first project the video features of length $T$ into a hidden dimension of $D$, then for each video encoder layer, we split the input video feature sequence into non-overlapping temporal windows of length $M$, resulting in the intra-window video representations $F^w \in \mathbb{R}^{K \times M \times D}$, where $K$ is the total number of the temporal windows. For the video features in the $i$-th window, i.e., $F_i^w \in \mathbb{R}^{M \times D}$, we first encode the detailed local information by performing temporal self-attention within the scope of that window as follows:

$$F_i^\ell = \text{Self-Attention}\left(F_i^w, F_i^w, F_i^w\right) \tag{1}$$

where $F_i^\ell \in \mathbb{R}^{M \times D}$ is the encoded video features with rich local contexts. Based on the local video features, we further apply a global-level self-attention on the global window features to connect different local contexts. Instead of using a simple pooling layer to aggregate the local window features, we exploit an attention-based method similar to the attention pooling in CLIP [58] to dynamically gather important local contexts for global interaction as:

$$F_i^g = \text{Cross-Attention}\left(\text{Avg}\left(F_i^l\right), F_i^l, F_i^l\right) \quad (2)$$

where $F^g \in \mathbb{R}^{K \times D}$ represents the global window features across the entire video, and $\text{Avg}(\cdot)$ is an average operation across temporal dimension for the local window features. Then the global window features are interacted with each other by a self-attention as:

$$F^g \leftarrow \text{Self-Attention}\left(F^g, F^g, F^g\right) \quad (3)$$

Next, we merge the information from the local and global contexts of the intra-window and inter-window features as:

$$F^V = \text{FFN}\left(\text{LN}\left(\text{Flatten}\left(F^\ell + \text{Rep}\left(F^g\right)\right)\right)\right) \quad (4)$$

where $\text{Rep}(\cdot)$ and $\text{Flatten}(\cdot)$ respectively indicate repeating the global window features by $M$ times in its corresponding window and unfolding the window-level representations into a feature sequence. $\text{LN}(\cdot)$ and $\text{FFN}(\cdot)$ denote the layer normalization operation and feed-forward network, respectively. $F^V \in \mathbb{R}^{T \times D}$ denotes the output features of a video encoder layer, and the output of each former layer will be further fed to the next layer for encoding.

### 4.3 Local-Global Iterative Decoder

Existing methods developed for single-paragraph video grounding [5, 63, 71] either encode the language query into a single global embedding [63] causing too much information loss, or employ self-attention on the complete multimodal sequence of all text features and video features [71], which incurs prohibitive resource cost thus is unsuitable in the multi-paragraph scenario. In this work, we explore a novel way to model the local-global query structures and cross-modal correspondences by iteratively reasoning about the local subparagraph features and global paragraph features.

To begin with, we first utilize a pre-trained RoBERTa [50] model to obtain the token-level language features from the $i$-th input paragraph, i.e., $F_i^L \in \mathbb{R}^{N_i^L \times D}$, where $N_i^L$ is the total number of language tokens in the $i$-th input paragraph. Afterwards, we jointly utilize a paragraph encoder and a subparagraph extractor to efficiently model the intrinsic local and global structures of the long text inputs, as shown in Figure 4. Concretely, we first embed all the token-level features within a paragraph into a global query feature $Q_i^g \in \mathbb{R}^D$ by an average-pooling operation. Then, we exploit $E$ learnable vectors $O^S \in \mathbb{R}^{E \times D}$ to extract the important subparagraph representations under the semantic guidance of $Q_i^g$ as follows:

$$Q_i^\ell = \text{Cross-Attention}\left(\text{LN}\left((Q_i^g W_1 + O^S W_2)\right), F_i^L, F_i^L\right) \quad (5)$$

where $W_1 \in \mathbb{R}^{D \times D}$ and $W_2 \in \mathbb{R}^{D \times D}$ are learnable projection matrices and $\text{LN}(\cdot)$ is the layer normalization operation. $Q_i^\ell \in \mathbb{R}^{E \times D}$ is the extracted subparagraph features that can be adaptively learned to represent meaningful local semantics specific to each paragraph for enhancing the cross-modal reasoning abilities within the paragraphs. Note that the number of extracted subparagraph features is typically small and the computation process will be efficient.

After obtaining the local subparagraph features and global paragraph features, we construct an iterative local-global reasoning process where each iteration involves intra-level reasoning, cross-modal reasoning and cross-level reasoning. Firstly, we conduct intra-level reasoning by employing two self-attention layers respectively within each window of local queries $Q_i^\ell$ and within all global queries $Q^g$. Afterwards, we achieve cross-modal reasoning

by extracting relevant information from $F^V$ to $Q^\ell$ and $Q^g$ by cross-attention layers. Then, we conduct cross-level reasoning also by two cross-attention layers, i.e., one is for extracting information from $Q^g$ to $Q^\ell$ and the other one is for extracting information from a window of local queries $Q_i^\ell$ to the corresponding global query $Q_i^g$. The cross-interacted features will serve as the output features of each decoder layer and are fed to the next layer for iterative decoding. Finally, the local and global output features, i.e., $Q_i^\ell$ and $Q_i^g$ of the last decoder layer are concatenated and fed to an MLP predictor to obtain the central timestamp $\hat{t}_c^i$ and duration $\Delta \hat{t}^i$ of the target interval for the $i$-th paragraph query. Then the temporal boundaries $\hat{\mathcal{T}}_i = (\hat{t}_s^i, \hat{t}_e^i)$ can be calculated as $\hat{t}_s^i = \hat{t}_c^i - \frac{\Delta \hat{t}^i}{2}, \hat{t}_e^i = \hat{t}_c^i + \frac{\Delta \hat{t}^i}{2}$.

### 4.4 Model Training

We train our model with a localization loss $\mathcal{L}_{loc}$ and an attention loss $\mathcal{L}_{att}$ which are formulated as follows:

$$\mathcal{L}_{loc} = \frac{1}{N} \sum_{i=1}^{N} \left[ \frac{1}{\lambda_1} \mathcal{L}_{l1}(\hat{\mathcal{T}}_i, \mathcal{T}_i) + \mathcal{L}_{GIoU}(\hat{\mathcal{T}}_i, \mathcal{T}_i) \right], \quad (6)$$

$$\mathcal{L}_{att} = -\frac{1}{N} \sum_{i=1}^{N} \log\left( \sum_{j=1}^{T} m_{ij} \cdot a_{ij} \right) \quad (7)$$

where $\mathcal{L}_{l1}$ and $\mathcal{L}_{GIoU}$ are L1 and GIoU [61] losses, respectively. $\hat{\mathcal{T}}_i$ is the predicted time span for the $i$-th query and $\mathcal{T}_i$ is the ground-truth. $\mathcal{L}_{att}$ is an attention loss on the global query features. The term $a_{ij}$ indicates the attention weights between the $i$-th global query feature and the $j$-th video feature, while $m_{ij}$ is an indicator that takes 1 if the $j$-th video feature is inside the ground-truth interval of the $i$-th query, and 0 otherwise. This loss explicitly encourages the model to learn higher attention weights between text queries and visual elements that are correlated. In total, the training loss is defined as the weighted sum of the above two losses as $\mathcal{L} = \lambda_1 \mathcal{L}_{loc} + \lambda_2 \mathcal{L}_{att}$, where $\lambda_1$ and $\lambda_2$ are the hyper-parameters to balance these two different kinds of losses.

## 5 Experiments

In this section, we illustrate our experimental setup and results for verifying and analyzing the effectiveness of our proposed method.

### 5.1 Experimental Setup

**Evaluation metrics.** For each query in the synopsis, we calculate the temporal Intersection over Union (IoU) between the predicted time span $\left[\hat{t}_s, \hat{t}_e\right]$ and the ground-truth time span $[t_s, t_e]$. Following previous video grounding methods [2, 23], we adopt two kinds of metrics to evaluate the performance: 1) mean IoU (mIoU) metric: average temporal IoU score calculated over all queries in the dataset; 2) IoU@$\theta$ metric: the proportion of queries with a temporal IoU score higher than $\theta$, here we use $\theta \in \{0.3, 0.5, 0.7\}$.

**Implementation details.** Our proposed method is implemented by PyTorch. The pre-extracted SlowFast, CLIP, and OCR features are aligned at sequence dimension and concatenated at channel dimension as the video feature input. We use a pre-trained RoBERTa [50] model to extract OCR features at each timestamp. The loss weights are set as $\lambda_1 = 1, \lambda_2 = 0.2$. For data augmentation, we choose to randomly shuffle the order of paragraphs in the same synopsis by a probability of $p$ during training and $p = \max(0, 1 - \frac{T_i}{T_{max}})$, where

**Table 4: Comparison results with state-of-the-art methods on multi-paragraph video grounding in SynopGround.**

| Method | mIoU | IoU@0.3 | IoU@0.5 | IoU@0.7 |
|--------|------|---------|---------|---------|
| Human | 85.1 | 97.3 | 92.5 | 85.0 |
| Random | 7.3 | 8.3 | 3.2 | 0.8 |
| DepNet [5] | 30.7 | 47.2 | 28.7 | 12.8 |
| PRVG [63] | 34.7 | 52.7 | 29.3 | 10.5 |
| LGMR (Ours) | **44.4** | **67.9** | **46.7** | **21.8** |

**Table 5: Evaluation on the effect of different features.**

| SlowFast | CLIP | OCR | mIoU | IoU@0.3 | IoU@0.5 | IoU@0.7 |
|----------|------|-----|------|---------|---------|---------|
| ✓ | × | × | 39.1 | 60.5 | 39.3 | 17.1 |
| × | ✓ | × | 39.8 | 61.7 | 40.0 | 16.5 |
| × | × | ✓ | 41.7 | 64.3 | 43.4 | 17.7 |
| ✓ | ✓ | × | 41.0 | 62.8 | 41.7 | 18.1 |
| ✓ | × | ✓ | 43.0 | 67.4 | 43.8 | 19.4 |
| × | ✓ | ✓ | 43.8 | 67.1 | 46.0 | 21.3 |
| ✓ | ✓ | ✓ | **44.4** | **67.9** | **46.7** | **21.8** |

$T_i$ is the index of the current training epoch and $T_{max}$ is set to 20. We adopt a local window length $M$ of 25 for our video encoder. The number of layers for video encoder and query decoder are set as 2 and 3, respectively. Our model is trained on 4 NVIDIA Tesla V100 GPUs by Adam [35] optimizer using a learning rate of 0.0001 and batch size of 16 for a total of 50 epochs within one day.

## 5.2 Experimental Results

**Performance Comparison.** As shown in Table 4, we evaluate the performance of our proposed LGMR on the challenging multi-paragraph video grounding task and compare it with the existing state-of-the-art methods DepNet [5] and PRVG [63]. DepNet is the baseline method proposed for multi-sentence video grounding, and PRVG is a concise and effective method based on DETR-like architectures [8]. For a fair comparison, all reported methods employ the same features as ours. The comparison results in Table 4 demonstrate that our proposed model achieves the best performance and outperforms others by a significant margin, which validates the effectiveness of the proposed LGMR method for addressing MPVG.

**Impact of different features.** To investigate the effect of different features, we conduct experiments with various combinations of SlowFast, CLIP, and OCR features. As shown in Table 5, we observe that using a single kind of features already yields satisfactory performance. Specifically, using the SlowFast, CLIP, or OCR features alone is able to produce an mIoU of 39.1%, 39.8%, and 41.7%, respectively. We notice that the CLIP features and OCR features are more helpful than the SlowFast features, which might be because 1) CLIP is pre-trained on large-scale image-text pairs, which makes it generalize better to the downstream task of video-language grounding. 2) The OCR features encode rich character-related and dialogue information, which is important for understanding the story plots in the narrative video. Additionally, we can see that the model using all three features together achieves the best performance on all metrics, showing that different kinds of features convey complementary information of the video content for language grounding.

**Table 6: Effect of the local-level modeling, global-level modeling, and cross-level reasoning in the iterative query decoder.**

| Local | Global | Cross | mIoU | IoU@0.3 | IoU@0.5 | IoU@0.7 |
|-------|--------|-------|------|---------|---------|---------|
| ✓ | × | × | 34.5 | 53.1 | 32.4 | 13.5 |
| ✓ | ✓ | × | 42.8 | 66.6 | 44.2 | 19.0 |
| ✓ | ✓ | ✓ | **44.4** | **67.9** | **46.7** | **21.8** |

**Table 7: Ablation studies on the proposed model designs. The GFLOPs measures computation complexity of the encoder.**

| Encoder | mIoU | GFLOPs | IoU@0.5 | IoU@0.7 |
|---------|------|--------|---------|---------|
| Vanilla Full | 42.5 | 12.6 | 45.1 | 20.5 |
| Local-Global | **44.4** | **9.6** | **46.7** | **21.8** |
| Loss | mIoU | IoU@0.3 | IoU@0.5 | IoU@0.7 |
| $\mathcal{L}_{loc}$ | 26.7 | 40.2 | 16.6 | 4.8 |
| $\mathcal{L}_{loc}$ and $\mathcal{L}_{att}$ | **44.4** | **67.9** | **46.7** | **21.8** |

**Effect of the local-global query modeling.** As shown in Table 6, we conduct detailed experiments to verify our proposed idea to model and reason the local-global structures of long queries. First, the model using only local queries for the cross-modal decoding process achieves a significantly lower performance compared to our final model. The reason is that only considering intra-query semantics neglects the rich contextual relationships among multiple correlated queries, while understanding the contexts is crucial for the multi-paragraph video grounding problem. Secondly, we observe significant gains in performance when jointly modeling the local and global structures of the long text inputs during decoding, showing the importance of our local-global query modeling.

**Ablation studies on design choices.** To further validate the rationality of our proposed model, we conduct ablation experiments on the designs of local-global temporal attention and cross-modal attention loss, as shown in Table 7. Firstly, we compare our model performance with that of a variant model where the local-global encoder is replaced by a vanilla full attention encoder [73]. The result suggests that our local-global encoder performs better in both accuracy and efficiency for long video inputs. Besides, we remove $\mathcal{L}_{att}$ and observe severe degradation in model performance. This highlights the importance of explicitly guiding the model to associate and align correlated visual and textual features.

## 6 Conclusion

In this work, we present a large-scale dataset for video-language grounding called SynopGround, which consists of over 2800 hours of long narrative videos with human-written synopses and manually annotated timestamps. It is the first video grounding dataset considering both long-form videos and long-text queries, and contains query descriptions conveying both low-level events as well as high-level plots for learning more complex and abstract concepts. We further introduce a challenging Multi-Paragraph Video Grounding (MPVG) task which incorporates long paragraph queries into multi-query video grounding. In addition, we propose a novel Local-Global Multimodal Reasoner (LGMR) to explicitly model the local-global structures of long-term inputs and conduct iterative reasoning within and across the two levels of structures, which can serve as a good starting point to inspire future research.

## Acknowledgments

This work was supported partially by the NSFC (U21A20471, U22A2095, 62076260, 61772570), Guangdong Natural Science Funds Project (2023B1515040025), Guangdong NSF for Distinguished Young Scholar (2022B1515020009), Guangdong Provincial Key Laboratory of Information Security Technology (2023B1212060026), and Guangzhou Science and Technology Plan Project (202201011134).

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
