# OpenReview forum: "SynopGround: A Large-Scale Dataset for Multi-Paragraph Video Grounding from TV Dramas and Synopses"
_acmmm.org/ACMMM/2024/Conference — MM2024 Poster_

### Official Review · Reviewer_TQPG · 2024-05-22

**Rating:** 2
**Confidence:** 4

**Summary:**

This paper proposed a large-scale multi-paragraph video grounding dataset which contains over 2800 hours TV show videos. This long paragraph query annotation requires method to model long range dependencies. Besides, it proposed a LGMR model to explicitly model local-global content in multimodal.

**Strengths:**

This paper is well written and clearly presented. The SynopGround dataset has longer videos and queries than existing video grounding dataset. The paragraph has both abstract expressions summarizing video storylines and specific descriptions portraying event details, involving multi-level semantics in video understanding. It proposed a baseline model that solves local and global semantics and achieves effectiveness in multi paragraph video grounding setting.

**Limitations:**

1.	The setting proposed by this paper is not challenging enough, since many works have focused on video paragraph grounding, i.e. giving a paragraph but localizing moments for each sentence in given paragraph. (i) The ground truth moments in old paragraph grounding setting have much shorter normalized temporal span than your setting, so it is more difficult to localize a very short moment other than localize a long paragraph. (ii) if many sentences are correctly localized in a paragraph, the paragraph then can be localized. Understanding the contextual correlations of a sentence and it belonged paragraph is more difficult than only understanding the paragraph. So your setting is not so challenging.
2.	The TV show have subtitles and the paragraphs annotations are mainly correlated to subtitles especially the high-level semantics, so a nlp method can solve it. The visual information contributes little. It does not encourage models to learn multimodal.

**Suitability:**

3

---

### Official Review · Reviewer_kiHF · 2024-05-24

**Rating:** 5
**Confidence:** 3

**Summary:**

This paper proposes a novel dataset for multi-paragraph video grounding from tv dramas and synopses. Different from existing datasets that focuses on short descriptions, the queries in this dataset are from synopses which are more detailed and challenging. The authors also propose a Local-Global Multimodal Reasoner method for this task. Extensive experiments show the proposed methods are effective.

**Strengths:**

- The proposed dataset and task are novel for video grounding tasks.
- The paper is well written and clear to understand.
- The proposed method is novel and specifically designed for long video grounding.

**Limitations:**

- Although the task proposed in the paper is multi-paragraph video grounding, I understand that it can also be applied to single-query scenarios. Adding some baselines for single-query video grounding, especially focusing on long video grounding (referring to the papers listed below), can make the experiments more comprehensive.

RGNet: A Unified Retrieval and Grounding Network for Long Videos.

Localizing Moments in Long Video Via Multimodal Guidance.

Cone: An efficient coarse-to-fine alignment framework for long video temporal grounding.

**Suitability:**

3

---

### Official Review · Reviewer_78Xu · 2024-05-24

**Rating:** 3
**Confidence:** 2

**Summary:**

This paper proposes a SynopGround dataset, a large-scale video grounding dataset. Based on the dataset, propose an MPVG task aimed at multi-paragraph video grounding. At the same time, an LGMR architecture is proposed; combining the two features of local and global to enhance the robustness of the model.

**Strengths:**

The strengths of this paper include:
1. Novelty: The paper introduces a dataset (SynopGround), a task (MPVG), and a model (LGMR).
2. Methodology: The authors propose a Local-Global Multimodal Reasoner model that includes a local-global temporal encoder and a local-global iterative decoder. This technique demonstrates promising potential in the task of MPVG.
3. Paper drawings look better.
4. Comprehensive Evaluation: The paper provides a rigorous experimental evaluation of the proposed model against strong baselines. This provides credible evidence of the model's effectiveness, offering a solid foundation for future improvement.

**Limitations:**

Review and Evaluation of Weaknesses:
1. Dataset construction: The dataset is manually annotated. How to ensure the balance of the data? How are data labeling rules designed? Is the synopsis data handwritten by humans? How to ensure the dataset quality?
2. Task definition: What is the difference between multi-paragraph video grounding (L328) and multi-sentence video grounding (L321)? What is the difference between one-time multi-paragraph video grounding and multiple one-paragraph video grounding?
3. In this way, what the MPVG task realistic application is? The significance and application of the MPVG task is not clear.
4. What do (d) and (e) in Figure 3 mean? unclear expression.
5. Unclear definition: Are Q_1, Q_2,...Q_N in the proposed method LGMR (Figure 4) continuous? What is the relationship between the corresponding V and Q? What is V, is a long video?
6. L560-L562, it seems that the training set, validation set, and test set are not completely separated.
7. Typo errors: e.g. (1) L621'figure 4'. (2)  'an' mIoU.
8. L856 ‘Specifically, using the SlowFast, CLIP, or OCR features alone is able to produce an mIoU of 39.1%, 39.8%, and 42.7%,’， ‘42.7%’ means?
9. How is the loss parameter lambda (lambda_loc and lambda_att) of L805 determined? Are there any other settings for comparison?
10. Why do some formulas use ‘=’ and some use ‘<-’? Does it have any special meaning?
11. There is no statement about publicly releasing the dataset and code of this method, which makes future re-production of the reported results a really challenging (if not impossible) task.

**Suitability:**

3

---

### Official Review · Reviewer_BVtg · 2024-05-25

**Rating:** 4
**Confidence:** 3

**Summary:**

This work proposes a new video-grounding dataset, SynopGround. Unlike previous datasets in this domain, SynopGround consists of long paragraph queries mapped to a segment of the respective video. These long queries reduce the ambiguity for possible one-to-many mapping for query and video segments. The data is collected from the online platform- Tencent Video, with proper acknowledgement and permissions. This work includes a handcrafted train-val-test split. Each video (sample) is an episode from a TV drama, and authors make sure that ~50% of the samples in the test/val set do not belong to the train set. The annotation process is explained well where annotators attain an 85% agreement score (IoU over individual's annotations). Despite proper data collection permissions, authors acknowledge the copyright restrictions and inability to provide the raw video frames. They assure only to share the pre-extracted features for reproducibility and enable future research over the shared features. Along with this dataset contribution, this paper presents a novel baseline: Long-Global Multimodal Resasoner (LGMR). The model is well explained and includes the implementation details, enough to replicate the work. The architecture design choices, modality selection, and loss function selection are justified by appropriate ablation studies.

**Strengths:**

1. Sample real-world use cases were provided where SynopGround can be utilised.
2. SynopGround dataset can support long-term textual video grounding research with complex queries.
3. Mutual agreement among the annotators, based on IoU of time-segment annotations, is claimed to be near 85%, which shows high agreement among the annotators.
4. Fine details about the novel Long-Global Multimodal Reasoner (LGMR) model are provided.
5. Sufficient experiments and ablation studies are conducted to justify the design choices behind the proposed LGMR model.
6. This submission includes detailed supplementary material with additional details about LGMR implementation, additional qualitative analysis with attention weights, and some visual results.
7. This submission contains good-quality plots and figures.

**Limitations:**

1. In Line 579, the authors mention that they will provide the pre-extracted features for public release and that, due to copyright restrictions, they will not be able to release the raw video frames. Public URLs will be provided. This restricts the research in the future because if the URLs get updated, the videos will be lost, and the mapping of the current timestamp annotations may not match with another version of the same video. Moreover, there may not be any way to extract video features from another model.
2. Line 573, How did authors extract subtitles via OCR? Subtitles are supposed to be extracted from audio transcriptions? Did the videos include embedded subtitles over frames?
3. I encourage authors to evaluate LGMR over other datasets, such as MAD, for completeness. Does it claim SoTA over other benchmarks? The results may boost the contribution or reveal some shortcomings of LGMR architecture for shorter queries.
4. The details about human evaluation are missing in Table 4. Please share details on how the human evaluation is conducted. How was the agreement among the annotators?

Minor concern
5. The task of Multi-Paragraph Video Grounding (MPVG) seems a special case of multi-sentence video grounding. Authors acknowledge this, so I suggest not emphasising this as a new task but building over an existing one.

**Suitability:**

3

---

### Meta-Review · Area_Chair_71T6 · 2024-07-07

**Recommendation:** Accept (Poster)
**Confidence:** 5

**Metareview:**

This paper introduces a large-scale video grounding dataset for multi-paragraph video grounding, and a strong baseline model for multi-paragraph video grounding is proposed by modeling the local-global information in the long video and paragraphs. The contribution of building such a dataset is solid and the baseline model also makes sense, which could be interesting to the research community on video grounding. The author also did a good rebuttal, addressing the major concerns from the reviewers, despite some remaining concerns regarding how to ensure the dataset quality with 80 annotators, comparing the normalized span with MAD dataset. The AC urges that the authors should revise the manuscript according to the reviewers' feedback and address all these remaining concerns. Meanwhile, the authors should also release the dataset, code and baseline models as they promise in the paper.